

# The short-term combined effects of temperature and organic matter enrichment on permeable coral reef carbonate sediment metabolism and dissolution

Coulson A. Lantz[1], Kai G. Schulz[1], Laura Stoltenberg[1], Bradley D. Eyre[1]

[1]Centre for Coastal Biogeochemistry, School of Environment, Science, and Engineering, Military Road

Southern Cross University Lismore 2480 NSW Australia

*Correspondence to:* Coulson A. Lantz (Coulsonlantz@gmail.com)



**Abstract**
Rates of gross primary production (GPP), respiration (R), and net calcification ($G_{net}$) in coral reef sediments are
expected to change in response to global warming (and the consequent increase in sea surface temperature) and
coastal eutrophication (and the subsequent increase in the concentration of organic matter (OM) being filtered
by permeable coral reef carbonate sediments). To date, no studies have examined the combined effect of
seawater warming and OM enrichment on coral reef carbonate sediment metabolism and dissolution. This study
used 22-hour *in situ* benthic chamber incubations to examine the combined effect of temperature (T) and OM, in
the form of coral mucus and phytodetritus, on GPP, R, and $G_{net}$ in the permeable coral reef carbonate sediments
of Heron Island lagoon, Australia. Compared to control incubations, both warming (+2.4 ℃) and OM increased
R and GPP. Under warmed conditions, R was enhanced to a greater extent than GPP, resulting in a shift to net
heterotrophy and net dissolution. Under both phytodetritus and coral mucus treatments, GPP was enhanced to a
greater extent than R, resulting in a net increase in GPP/R and $G_{net}$. The combined effect of warming and OM
enhanced R and GPP, but the net effect on GPP/R and $G_{net}$ was not significantly different from control
incubations. These findings show that a shift to net heterotrophy and dissolution due to short-term increases in
seawater warming may be countered by a net increase GPP/R and $G_{net}$ due to short-term increases in nutrient
release from OM.
**1. Introduction**
Despite occupying only 7.5% of the seafloor, coastal marine sediments are responsible for a large fraction
(55%) of global sediment organic matter oxidation (Middelburg et al., 1997). Of the coastal marine sediment
environments, coral reef sediments are one of the most severely threatened by global climate change (Halpern et
al., 2007). Rates of sediment autotrophic production (gross primary productivity; GPP) on coral reefs are
generally greater than rates of heterotrophic metabolism (respiration; R) (GPP/R > 1), such that the sediments
are generally a net source of oxygen (Atkinson, 2011). Similarly, rates of sediment calcification are generally
greater than rates of sediment dissolution ($G_{net} > 0$) on most reefs under current ocean conditions, such that coral
reef sediments on 24-hour diel timescales are net precipitating, resulting in the long-term burial of carbon in the
form of calcium carbonate (Eyre et al., 2014; Andersson, 2015). This long-term production of calcium carbonate
is an important component of reef formation and the creation of sandy cays (Atkinson, 2011). However, due to
anthropogenically-mediated processes such as sea surface temperature (SST) warming (Levitus et al., 2000) and
coastal eutrophication (Fabricius, 2005), coral reefs sediments may soon be subjected elevated SSTs and excess



concentrations of OM (Rabouille et al., 2001). This could ultimately impact the balance in GPP/R and $G_{net}$ in the
sediment and potentially reduce the long-term accumulation of carbonate material on coral reefs (Orlando and
Yee, 2016).
Given the recent predictions of SST increases on coral reefs of between 1.2 to 3.2 ºC by the end of this century
(IPCC, 2013), there are concerns that the net benthic metabolic balance in coral reef sediments may shift away
from net production and net calcification to a state of net heterotrophy and net dissolution (Pandolfi et al., 2011).
While several coral reef studies have examined the response in individual calcifying organisms to increased
seawater temperature (T) (e.g., Johnson and Carpenter, 2012; Shaw et al., 2016), only one study (Trnovsky et
al., 2016) has examined the response in the permeable coral reef carbonate sediments. The majority of warming
studies on marine sediments have been performed *ex situ* in more pole-ward latitudes (temperate to arctic
environments) over a wide range of temperatures (2 – 30 ºC) (e.g., Tait and Schiel, 2013; Hancke et al., 2014).
The bacterial communities residing in marine sediments generally display a hyperbolic temperature-production
relationship where GPP increases with T (~ + 32 % per 1 ºC increase) until an optimal rate is reached roughly
+2 – 3 ºC above naturally observed seasonal maxima. This T-GPP relationship then declines at higher
temperatures (+4 - 6 ºC) due to the deactivation of component reactions (Bernacchi et al., 2001). In arctic and
temperate marine sediment communities, the increase in T can alter the balance between GPP and R, with an
observed shift towards net heterotrophy (GPP/R < 1) (e.g., Arnosti et al., 1998; Hancke and Glud, 2004; Weston
and Joye, 2005). Trnovsky et al. (2016) found that warming also decreased GPP/R in coral reef sediments and
reduced $G_{net}$ due to enhanced sediment dissolution.
Ultimately, the magnitude of potential shifts in coral reef sediment GPP/R and $G_{net}$ under global warming
scenarios will depend critically on the availability of organic matter (OM) substrate for remineralisation
(Ferguson et al., 2003; Rabalais et al., 2009). A review of coral reef sediment studies has shown that carbonate
sediment dissolution is strongly controlled by the extent of OM decomposition in the sediments (Andersson,
2015). Coral reefs are classically characterized as oligotrophic, relatively deficient in major inorganic nutrients
(Koop et al., 2001). Despite this classification, the relatively high rates of GPP (1 to 3 mol C $m^{-2}$ $d^{-1}$) for these
ecosystems (Odum and Odum, 1955), are evidence of the tightly coupled nutrient cycling between autotrophs
and heterotrophs. However, the balance in sediment metabolism on coral reefs may change in response to OM
over-enrichment associated with eutrophication (Bell, 1992). Coral reefs affected by eutrophication (e.g.,
Hawaii (Grigg, 1995), Indonesia (Edinger et al., 1998), Jamaica (Mallela and Perry, 2007), Puerto Rico (Diaz-
Ortega and Hernandez-Delgado, 2014)) all exhibit elevated concentrations of OM in the water column



(particulate OM: 10 – 50 μmol C L$^{-1}$) and above average rates of sedimentation (5 – 30 mg cm$^{-2}$ d$^{-1}$). Elevated
concentrations of OM and increased rates of terrestrially derived sedimentation on coral reefs can cause a
decline in hard coral cover and a relative increase in macroalgal cover, resulting in an overall degradation of
coral reef habitat (Fabricius, 2005).
Eutrophication can increase the amount of OM processed in coral reef sediments through several processes, two
of which were simulated in this study; 1) through local phytoplankton blooms in the water column in response
to the runoff of inorganic and organic nutrients and the eventual sediment deposition of dead phytoplankton
(referred to herein as phytodetritus)  (Furnas et al., 2005) and 2) the release of coral mucus into the reef water
column as a stress response of scleractinian corals to increased sedimentation and the subsequent sediment
deposition of this bacteria-rich protein matrix (Ducklow and Mitchell, 1979). The sediment deposition of OM
provides labile carbon substrate (and associated nitrogen and phosphorous) for immediate consumption by
autotrophic and heterotrophic bacterial communities.
Studies which have examined the effect of increased concentrations of OM, such as coral mucus (e.g., Wild et
al., 2004a) or coral spawn and phytodetritus (e.g., Eyre et al., 2008), on coral reef sediment metabolism have
shown a short-term increase in GPP/R, contrasting the results provided from short-term temperature studies on
coral reef sediments, where GPP/R decreased (Trnovsky et al., 2016). Experimental additions of coral mucus
from *Acropora* spp. on Heron Island, Australia (conducted only in the dark) induced a ~ 1.5-fold increase in R
(Wild et al., 2004b) while additions of *Fungia* spp. mucus from a reef in Aqaba, Jordan (also conducted in the
dark; Wild et al., 2005) showed a ~ 1.9-fold increase in R. OM associated with a mass coral spawning event
(coral gametes and subsequent phytodetritus produced in the water column) on Heron Island, Australia caused a
2.5-fold increase in sediment R and a 4-fold increase in sediment GPP (Glud et al., 2008).  Unlike the short-term
response in GPP/R to T, sediment metabolism remained net-autotrophic during the spawning event at Heron
Island, with GPP/R ratios rising as high as 2.5 – 3.0 (Glud et al., 2008), implying that nutrients recycled from
OM stimulated GPP in excess of R (Eyre et al., 2008) on relatively short timescales (hours to days). However,
studies which have examined the effect of excess OM on coral reef sediment metabolism over longer time scales
(weeks to months) have shown that, ultimately, GPP/R eventually shifts to net heterotrophy (e.g., Andersson,
2015; Yeakel et al., 2015). This suggests that despite an initial OM-induced increase in GPP/R, the net long-
term effect within reef sediments may be a preferentially heterotrophic recycling of nutrients released from
organic matter degradation. Therefore, questions remain if a predicted temperature-driven shift to net
heterotrophy will be exacerbated or mitigated by the presence of excess organic matter filtered by coral reef



sediments. There are, to date, no studies that have examined the effect of OM on coral reef sediment $G_{net}$. A
short-term increase in GPP/R in response to OM implies that sediment $G_{net}$ may be enhanced by excess
concentrations of OM given that coral reef sediments generally exhibits a positive GPP/R-$G_{net}$ relationship
(Cyronak et al., 2016) whereas a long-term decrease in GPP/R may result in a decrease in sediment $G_{net}$.
Therefore, seawater warming and eutrophication will likely increase GPP and R in coral reef sediments, but,
altogether, there is a lack of research demonstrating how these perturbations, specifically in combination, will
affect the balance in coral reef sediment organic (GPP/R) and inorganic ($G_{net}$) metabolism. To meet these needs,
this study performed incubations using benthic chambers placed *in situ* in a shallow coral reef sediment
environment for a period of 24 hours. Phytodetritus and coral mucus were added to chamber seawater under
ambient and increased SST (+2.4 °C) conditions and the corresponding changes in GPP, R, and $G_{net}$ were
measured. We hypothesized that the short-term combined treatments of seawater warming and OM loading
would enhance GPP and R in the sediment, but, given the previously shown short-term response in GPP/R and
$G_{net}$ to seawater warming (decrease in GPP/R and $G_{net}$) and net response to OM loading (decrease in GPP/R, $G_{net}$
response unknown), there would be a net decrease in GPP/R and $G_{net}$ relative to control treatments.
**2. Methods**
**2.1 Study site**
This study was conducted at Heron Island, Australia (23° 27'S, 151° 55'E) in November 2016. The island is
situated near the Tropic of Capricorn, at the southern end of the Great Barrier Reef (GBR) and contains a ~ 9 ha
island surrounded by a ~ 24 ha coral reef with an average hard coral cover of roughly 39% (Salmond et al.
2015). The study site was located on the leeward side of the reef flat, roughly 100 m from the island shore, in a
sandy patch where water depth varies between ~ 0.1 – 2.7 m due to semi-diurnal tidal changes. The site was
predominately covered in permeable $CaCO_3$ sediments (~ 63%) with interspersed patches of hard coral
dominated by *Acropora* spp. (Roelfsema et al., 2002). The $CaCO_3$ sediment at this site has a ~ 2:1 ratio of
aragonite: high magnesium calcite (Cyronak et al., 2013a). Sediment grain size: 12.1%. 2 mm, 30.5% between 1
and 2 mm, 27.3% between 500 mm and 1 mm, 14.1% between 250 mm and 500 mm, 11.2% between 125 mm
and 250 mm, 4.2% between 63 mm and 125 mm, and 0.6%, 63 mm (Cyronak et al., 2013b).
**2.2 Experimental design**



A total of four 22-hour diel incubations were conducted during 5 - 12 Nov 2016 in advective benthic chambers.
Benthic net primary production (NPP), gross primary productivity (GPP), respiration (R), and net calcification
($G_{net}$) were compared under ambient (~ 0.63 μmol C L$^{-1}$) and elevated concentrations of organic matter (OM)
(additions of ~ 21.3 μmol C L$^{-1}$ phytodetritus or ~ 23.6 μmol C L$^{-1}$ coral mucus) at ~ 28.2 ºC and ~ 30.6 ºC in
an orthogonal design. Eight chambers were used per incubation day, with each of the four OM-temperature
combinations replicated in two randomly assigned chambers (Fig. 1). The first two incubations included two
replicate chambers using phytodetritus crossed with temperature (6 and 7 Nov 2016) while the next two
incubations included two replicate chambers using coral mucus crossed with temperature (9 and 11 Nov 2016).
Incubations were started at sunset (18:00) and ended the following day at dusk (16:00). This allowed for a two-
hour period (16:00 – 18:00) where chambers could be moved to a new area of sediment, closed, and heated to
the desired temperature offset before beginning the next set of incubations.

### 2.3 Benthic chambers

Advective benthic chambers were constructed out of clear acrylic with a height of 33 cm and a diameter of 19
cm (Huettel and Gust, 1992). A motorized clear disc in the top of the chamber was programmed to spin at a rate
of 40 revolutions per minute, which had previously been determined to induce an advection rate of ~ 43 L m$^{-2}$ d$^{-1}$
$^{1}$ at the study site (Glud et al., 2008). Roughly 10 - 12 cm of the base of the chamber was inserted into the
sediment such that a ~ 4 L water column of seawater was enclosed within the chamber (height ~ 15 cm) upon
closing by the lid. The exact water volume varied within each chamber and was calculated for each incubation
by multiplying known areal coverage by measured chamber height (at three positions above the sediment). Prior
to closing the chambers, the tops were left open for ~ 1 hour to allow settlement of disturbed sediment.
Chambers were then sealed ~ 1 hour prior to the beginning of each incubation to allow each temperature
treatment chamber to reach the desired temperature offset. Following this, at the beginning of each incubation,
selected chambers (four of the eight) were injected with OM (either coral mucus or phytodetritus).

### 2.4 Temperature manipulation

The international panel on climate change (IPCC) representative concentration pathway (RCP) 8.5 projects an
average 2.2 - 2.7 ºC increase in SST (IPCC, 2013). A similar increase in temperature within the benthic
chambers was achieved with 5W, silicone-heating pads (RS Australia) inserted inside of each of the four
temperature treatment chambers (e.g., Trnovsky et al., 2016). These pads resided in the middle of the chamber
water column and were powered by a 12 V battery on a surface support station tethered roughly 3 m away.



Temperature and light was measured in all eight chambers and in the water column using HOBO temperature
loggers, which recorded temperature (ºC) and light (Lux) at an interval of fifteen minutes. Light intensity (Lux)
was converted to μmol quanta of photosynthetic active radiation (PAR) m$^{-2}$ s$^{-1}$ using a conversion factor of
0.0185, derived from correlations with PAR measurements of a calibrated ECO-PAR (Wetlabs) sensor over a
period of five days (R$^2$ = 0.89).
Heating pads increased temperature (T) within the chambers by 2.4 ± 0.5 ºC and maintained this offset on top of
the natural diel temperature fluctuations measured in the control chambers (Table 1). As HOBO temperature
loggers may record potentially higher than surrounding seawater temperatures due to internal heating of the
transparent plastic casing (Bahr et al., 2016; Trnovsky et al., 2016), HOBO temperature data was corrected for
precision (48-hour side-by-side logging of all nine loggers in an aquarium) and accuracy (deployment next to an
*in situ* SeapHOx (Sea-Bird Electronics) for 48 hours). The conductivity sensor of the SeapHOx was used to
record water column salinity for the duration of the experiment (7 days) at a sampling frequency of 30 minutes.
**2.5 Organic matter manipulations**
Phytodetritus (PD) was injected into treatment chambers to achieve a concentration increase by ~ 20 μmol C L$^{-}$
$^{1}$, a value analogous to mean conditions observed on degraded eutrophic coral reefs, where water column
conditions can range from 10 to 50 μmol C L$^{-1}$ (Fabricius et al., 2005, Diaz-Ortega and Hernandez-Delgado,
2014). Unfiltered seawater (6 L) was collected from the coastal ocean adjacent to the SCU laboratories (Lennox
Head, NSW, Australia), containing naturally occurring assemblages of phytoplankton species common to the
East Australian current. Collected seawater was stimulated with additions of 128 μmol L$^{-1}$ NO$_3$$^{-}$, 8 μmol L$^{-1}$
PO$_4$$^{3-,}$ and 128 μmol L$^{-1}$ H$_4$SiO$_4$ (buffered by additions of 256 μmol L-1 of HCl), and a solution of trace metals
and vitamins (F$_{1/8}$; Guillard, 1975). Total amounts of nutrients were chosen to allow for a community
production of up to 850 μmol C L$^{-1}$ assuming a classical C: N: P Redfield ratio of 116:16:1 and a N:Si
requirement of diatoms of 1. After a week of incubation at 150 μmol quanta of PAR m$^{-2}$ s$^{-1}$ at 20 ºC, the
phytoplankton community was concentrated to 1/50$^{th}$ the original volume (0.12 L) via gentle (> -0.2 bar)
vacuum filtration and rinsed with artificial seawater to remove residual concentrations of dissolved organic and
inorganic nutrients. The resulting phytoplankton concentrate (measured at 8.5 μmol C L$^{-1}$ and 0.9 μmol N L$^{-1}$ of
particulate organic carbon (POC) and nitrogen (PON), respectively; see section 2.6) was stored in the dark at 4.0
ºC until experimental use (6 days). At the beginning of an incubation, 10 ml of the dead phytoplankton
concentrate, referred to as PD, was injected into each treatment chamber (~4 L volume), raising the



concentration of carbon and nitrogen by ~ 21.3 ± 1.0 µmol C $L^{-1}$ and ~ 2.2 ± 0.8 µmol N $L^{-1}$, respectively (Table

183  1).

The amount of coral mucus (CM) added to the chambers was chosen to represent a reef-wide discharge of CM
based on reported average mucus secretion rates for *Acropora* spp. (4.8 L mucus $m^{-2}$ $d^{-1}$; Wild et al., 2004a), the
dominant genus on the Heron Island reef flat. Mucus was collected from scattered branching coral fragments
(*Acropora* spp.) using a non-destructive method whereby loose individual colonies naturally exposed to air
during low tide were inverted so that gravity facilitated the pooling of secreted mucus through a cone filter into
a large, 5 L beaker. This mucus was returned to the lab, particle filtered (5.0 µM) to remove the bulk of
seawater, re-filtered to separate out particle carbonates, and stored in the dark at 4.0 °C until experimental use (2
days). Ninety-four ml of mucus was injected into each treatment chamber to simulate the equivalent reported
*Acropora* spp. mucus secretion rate (4.8 L mucus $m^{-2}$ $d^{-1}$) for Heron Island given the average percent of this
secreted mucus filtered by the sand (~ 70%; Wild et al., 2004a) and the benthic area enclosed by each chamber
(0.028 $m^2$). Based on POC and PON concentrations (measured at 12.1 mmol C $L^{-1}$ and 0.8 mmol N $L^{-1}$,
respectively; see section 2.6) this represented an addition of ~ 23.6 ± 1.1 µmol C $L^{-1}$ and 1.4 ± 0.4 µmol N $L^{-1}$
(Table 1).
**2.6 Sample collection and analysis**
Seawater samples (120 ml total) were extracted from the top of each chamber via two two-port valves using two
60 ml syringes without headspace at ~12 hour intervals (sunset, dawn, and dusk) and returned to the lab for
immediate analysis and/or preservation. 10 ml of unfiltered seawater from each chamber was analysed for
dissolved oxygen (DO; mg $L^{-1}$) with a Hach HQ 30d meter and Luminescent DO (LDO) probe. Samples for
seawater total alkalinity ($A_T$; µmol $kg^{-1}$) were filtered (0.45 µm; Chanson and Millero, 2007) and stored in 100
ml plastic, airtight bottles for immediate analysis (< 24 hours). Samples for dissolved inorganic carbon (DIC;
µmol $kg^{-1}$) were also filtered (0.45 µM) into the bottom of 6 ml crimp vials (rubber butyl septum) with 5 ml
overflow, and poisoned (6 µl of saturated $HgCl_2$; Dickson, 2007).
Seawater $A_T$ was analysed using a potentiometric titration method (Dickson, 2007) on a Metrohm 888 Titrando
automatic titrator using ~ 10 ml of seawater per sample. DIC was analysed on a Marianda AIRICA coupled to a
Li-COR LI 7000 $CO_2/H_2O$ Analyzer using ~ 1.6 ml of seawater per sample whereby four replicates of 400 µl
were analysed for each sample and a best of three approach was used for each DIC calculation. $A_T$ and DIC
sample precision was estimated with replicate analyses conducted on every fifth sample ($A_T$ SE = ± 1.7 µmol



kg$^{-1}$; DIC SE = ± 1.8 µmol kg$^{-1}$). Measurements were corrected against certified reference material (CRM;
Batch 155) from the Scripps Institute of Oceanography (A$_T$ SE = ± 2.2 µmol kg$^{-1}$; DIC SE = ± 1.3 µmol kg$^{-1}$).
Parameters for the seawater carbonate system ($\Omega_{ar}$, pH$_T$ (total scale)) were calculated from measured A$_T$, DIC,
temperature, and salinity using the R package seacarb (Lavigne and Gattuso, 2013) with K$_1$ and K$_2$ constants
applied from Mehrbach et al. (1973) and refit by Dickson and Millero (1987). Because changes in A$_T$ could be
due to processes other than the precipitation and dissolution of carbonates (e.g., sulfate reduction associated
with organic matter additions), fluxes in DIC were corrected for assumed A$_T$ fluxes due to calcium carbonate
precipitation/dissolution (0.5 moles CO$_2$: 1 mole A$_T$) and compared against fluxes in O$_2$, with an expected 1:1
molar flux ratio (DIC$_{org}$ : O$_2$).
Prior to chamber additions subsamples (1 ml, n = 3) were taken from the concentrated PD culture, CM, and the
water column and analysed for particulate organic carbon (POC) and nitrogen (PON). These subsamples were
filtered on pre-combusted 25mm GF/F filters, dried at 60 °C, fumed with 12 M HCl to dissolve any particulate
carbonates on the filter, and wrapped in pre-combusted tin capsules. These capsules were analysed for carbon
(C) and nitrogen (N) using an elemental analyser (Thermo Flash ES) coupled to an isotope ratio mass
spectrometer (Thermo Delta V PLUS) via a Thermo Conflo V (see Eyre et al. 2016, for details).
**2.7 Calculating sediment metabolism**
Benthic metabolism (NPP, GPP, R, G$_{net}$) in each chamber was estimated based on the flux of measured solutes
(DO, and A$_T$, respectively). For flux calculations, DO was converted from mg L$^{-1}$ to mmol L$^{-1}$. A$_T$ and DIC were
converted from µmol kg$^{-1}$ to mmol L$^{-1}$ using calculated temperature and salinity dependent seawater density.
The solute flux equation (Glud et al., 2008) was as follows:
$Equation 1 : F = \frac{\Delta S \times v}{A \times \Delta t}$
Where $F$ (mmol m$^{-2}$ hr$^{-1}$) is the net flux in solute, $\Delta S$ (mmol L$^{-1}$) is the change in solute concentration, $v$ (L) is
the chamber volume, $A$ (m$^2$) is the area of sediment enclosed by the chamber, and $\Delta t$ (hours) is the time elapsed
between seawater samplings. Rates of sediment net primary production (NPP), gross primary production (GPP),
and respiration (R) were calculated from O$_2$ fluxes (mmol O$_2$ m$^{-2}$ hr$^{-1}$), and rates of net sediment calcification
(G$_{net}$) were calculated from A$_T$ fluxes (mmol CaCO$_3$ m$^{-2}$ hr$^{-1}$) (Table 2).
To determine the numerical relationship between a 10 °C change in temperature and GPP and R, Q$_{10}$ values
were estimated for temperature treatments according to the following equation:



$$Equation 2 : Q_{10} = \left(\frac{M2}{M1}\right)^{\left(\frac{10}{T_2 - T_1}\right)}$$
where $M1$ is the metabolic rate (GPP or R) at temperature $T_1$ (control) and $M2$ is the metabolic rate (GPP or R,
respectively) at temperature $T_2$ (warming treatment), with $T_1 < T_2$.
**2.8 Statistical analyses**
Results are displayed as the mean ± standard error (SE). Data was organized as the 22-hour average (diel) of day
and night values and was pooled together within each T, OM, and T + OM treatment. All statistical analyses
were performed with the SPSS statistics software (SPSS Inc. Version 22.0) running in a Windows PC
environment, and the assumptions of normality and equality of variance were evaluated with graphical analyses
of the residuals. To test for the effect of each treatment (T, PD, and CM) on respiration, photosynthesis, and
calcification, measured 22-hour R, NPP, GPP, and $G_{net}$ were analysed using a repeated-measures three-way
analysis of variance (ANOVA). In this model, temperature and OM (PD and CM) were fixed effects, the within-
subject factor was time (days), and replicate chambers were a nested effect. To compare the significance of
temperature and OM between and within treatment chambers, a one-way ANOVA model was used in which
chamber was the fixed effect and average seawater temperatures (ºC) and POC and PON concentrations,
respectively, were treated as the response variable. In these analyses, Bonferroni post-hoc test were used to
conduct pair-wise comparisons between treatments.
The variation in NPP, GPP, R, and $G_{net}$ provided the opportunity to explore the relationship between net organic
and inorganic metabolism across all treatments. A Pearson correlation was used to test for an association
between sediment $G_{net}$, NPP, GPP, R, and GPP/R. Where each of these pairwise correlations were statistically
significant, Model I regression techniques were used to fit a linear relationship for the purpose of predicting
inorganic metabolism ($G_{net}$) from organic metabolism (NPP, GPP, R, GPP/R).
**3. Results**
**3.1 Measured seawater chemistry and sediment metabolism in control chambers**
Temperature measured in the water column throughout the experiment (27.6 ± 1.3 ºC) exhibited typical diel
changes and was slightly cooler relative to the average temperature inside control chambers (-0.8 ± 0.5 ºC) (Fig.
2). Mean water column salinity throughout the experiment was 35.8 ± 0.1. Over the course of each diel
incubation period, changes in water chemistry (Fig. 3) were driven by benthic metabolism. Control (C)



chambers, pooled between all four incubations (n = 9), had an R of -1.3 ± 0.5 mmol $O_2$ $m^{-2}$ $hr^{-1}$ and an NPP of
1.9 ± 0.3 mmol $O_2$ $m^{-2}$ $hr^{-1}$, yielding a GPP of 3.2 ± 0.6 mmol $O_2$ $m^{-2}$ $hr^{-1}$ and a GPP/R of 1.31 ± 0.12. C
chambers were net dissolving at night (-0.9 ± 0.2 mmol $CaCO_3$ $m^{-2}$ $hr^{-1}$) and net calcifying during the day (1.3 ±
0.2 mmol $CaCO_3$ $m^{-2}$ $hr^{-1}$). Overall, 24-hour diel $G_{net}$ ($F_{1,31}$ = 122.82, p < 0.05) was net calcifying (0.2 ± 0.1
mmol $CaCO_3$ $m^{-2}$ $hr^{-1}$). Mean particulate organic carbon (POC) and nitrogen (PON) concentrations in the four C
chambers was 0.63 ± 0.1 µg C $L^{-1}$ and 0.12 ± 0.1 µg N $L^{-1}$, respectively.
**3.2 The effects of temperature on sediment metabolism**
Mean seawater temperature in the C and temperature (T) treatments during the four incubation periods was 28.2
± 1.1 ºC and 30.6 ± 1.2 ºC, respectively (Table 1). Temperature differed between C and T treatments ($F_{1,31}$ =
384.38, p < 0.05), but there was no significant difference between replicate chambers within each treatment
($F_{1,31}$ =0.76, p = 0.768). Temperature in all eight chambers exhibited typical diel changes throughout all four
incubation periods, driven by sunlight and tidal changes in water depth (Fig. 2). Treatment chambers followed
the same natural diel change measured in control chambers and maintained an average +2.4 ± 0.5 ºC offset over
the course of the study (Table 1).
Within the T treatments there was no significant difference in estimated metabolic rates between all four
incubations ($F_{1,31}$ =1.2, p = 0.238), so rates were pooled. During the fourth incubation, one T treatment was lost
due to a broken heater and this chamber was treated as a third control replicate. Seawater warming increased R
to -3.5 ± 0.4 mmol $O_2$ $m^{-2}$ $hr^{-1}$ ($F_{1,31}$ = 260.38, p < 0.05) (Table 3), NPP to 2.9 ± 0.4 mmol $O_2$ $m^{-2}$ $hr^{-1}$ ($F_{1,31}$ =
192.17, p < 0.05), and GPP to 6.4 ± 0.5 mmol $O_2$ $m^{-2}$ $hr^{-1}$ ($F_{1,31}$ = 160.61, p < 0.05) (Fig. 4). Overall, warming
decreased GPP/R to 0.93 ± 0.05 ($F_{1,31}$ = 79.02, p < 0.05), indicating a shift from net autotrophy to net
heterotrophy (Fig. 5). Mean calculated $Q_{10}$ values, averaged across T treatments from all four incubations, were
10.7 ± 3.1 for R and 7.3 ± 1.2 for GPP. Warmed chambers were net dissolving at night (-1.9 ± 0.2 mmol $CaCO_3$
$m^{-2}$ $hr^{-1}$) and net calcifying during the day (1.7 ± 0.2 mmol $CaCO_3$ $m^{-2}$ $hr^{-1}$). Overall, warming decreased $G_{net}$ to
-0.2 ± 0.1 mmol $CaCO_3$ $m^{-2}$ $hr^{-1}$ ($F_{1, 31}$ = 122.82, p < 0.05) (Fig. 6), indicating a shift to net dissolution.
**3.3 The effects of organic matter on sediment metabolism**
Mean POC and PON concentrations in the four phytodetritus (PD) treatment chambers was 21.7 ± 1.0 µg C $L^{-1}$
and 2.3 ± 0.8 µg N $L^{-1}$, respectively (POC:PON ~ 9:1) (Table 1). During the PD treatment incubations, there
was no significant difference in metabolic rates between incubations ($F_{1,15}$ = 0.32, p = 0.299), so estimated
metabolic rates were pooled within the PD-only treatments. PD increased R to -2.6 ± 0.5 mmol $O_2$ $m^{-2}$ $hr^{-1}$ ($F_{1,15}$



$= 16.77$, $p < 0.05$) and increased NPP to $5.3 \pm 0.5$ mmol $O_2$ m$^{-2}$ hr$^{-1}$ ($F_{1,15} = 245.86$, $p < 0.05$), thereby increasing
GPP to $7.9 \pm 0.4$ mmol $O_2$ m$^{-2}$ hr$^{-1}$ ($F_{1,15} = 212.64$, $p < 0.05$) and increasing GPP/R to $1.54 \pm 0.11$ ($F_{1,15} = 13.92$,
$p < 0.05$). Chambers treated with PD were net dissolving at night ($-1.5 \pm 0.2$ mmol $CaCO_3$ m$^{-2}$ hr$^{-1}$) and net
calcifying during the day ($2.8 \pm 0.3$ mmol $CaCO_3$ m$^{-2}$ hr$^{-1}$). Overall, PD increased $G_{net}$ to $0.6 \pm 0.2$ mmol $CaCO_3$
m$^{-2}$ hr$^{-1}$ ($F_{1,15} = 134.27$, $p < 0.001$).
Mean POC and PON concentrations in the four coral mucus (CM) treatment chambers was $24.2 \pm 1.1$ µg C L$^{-1}$
and $1.5 \pm 0.4$ µg N L$^{-1}$, respectively (POC:PON ratio ~ 16:1). During CM incubations, there was no significant
difference in metabolic rates between incubations ($F_{1,15} = 0.42$, $p = 0.448$), so estimated metabolic rates were
pooled together within CM-only treatments. CM increased R to $-2.0 \pm 0.4$ mmol $O_2$ m$^{-2}$ hr$^{-1}$ ($F_{1,15} = 7.34$, $p <$
$0.05$) and increased NPP to $4.4 \pm 0.5$ mmol $O_2$ m$^{-2}$ hr$^{-1}$ ($F_{1,15} = 134.51$, $p < 0.05$), thereby increasing to $6.4 \pm 0.6$
mmol $O_2$ m$^{-2}$ hr$^{-1}$ GPP ($F_{1,15} = 99.24$, $p < 0.05$) and increasing GPP/R to $1.61 \pm 0.2$ ($F_{1,15} = 34.17$, $p < 0.05$).
Chambers treated with CM were net dissolving at night ($-1.3 \pm 0.2$ mmol $CaCO_3$ m$^{-2}$ hr$^{-1}$) and net calcifying
during the day ($2.4 \pm 0.3$ mmol $CaCO_3$ m$^{-2}$ hr$^{-1}$). Overall, CM increased $G_{net}$ to $0.5 \pm 0.2$ mmol $CaCO_3$ m$^{-2}$ hr$^{-1}$
($F_{2,22} = 100.61$, $p < 0.05$).

**3.4 The combined effects of temperature and organic matter on sediment metabolism**

In the first two incubations (T + PD), there was no significant difference in metabolic rates between days ($F_{1,15}$
$= 1.23$, $p = 0.135$), so estimated metabolic rates were pooled together within the T + PD treatments. T + PD
increased R to $-3.1 \pm 0.5$ mmol $O_2$ m$^{-2}$ hr$^{-1}$ ($F_{1,15} = 46.4$ $p < 0.001$), increased NPP to $4.7 \pm 0.5$ mmol $O_2$ m$^{-2}$ hr$^{-1}$
($F_{1,15} = 16.31$, $p < 0.05$), and increased GPP to $7.8 \pm 0.5$ mmol $O_2$ m$^{-2}$ hr$^{-1}$ ($F_{1,15} = 8.81$, $p < 0.05$). GPP/R in T +
PD treatments was $1.27 \pm 0.18$ (PD+T), a change that was not significantly different from control chambers
($F_{1,15} = 2.75$, $p = 0.122$). Chambers treated with T + PD were net dissolving at night ($-1.9 \pm 0.2$ mmol $CaCO_3$ m$^{-2}$
hr$^{-1}$) and net calcifying during the day ($2.6 \pm 0.3$ mmol $CaCO_3$ m$^{-2}$ hr$^{-1}$). Overall, 22-hour diel $G_{net}$ in T + PD
treatments was $0.3 \pm 0.2$ mmol $CaCO_3$ m$^{-2}$ hr$^{-1}$, a change which was not significantly different from control
chambers ($F_{1,15} = 0.70$, $p = 0.417$).
In the two last incubations (T + CM), there was no significant difference in metabolic rates between days ($F_{1,15}$
$= 1.73$, $p = 0.110$), so estimated metabolic rates were pooled together within the combined T + CM treatments. T
+ CM and increased R to $-2.9 \pm 0.4$ mmol $O_2$ m$^{-2}$ hr$^{-1}$ ($F_{1,15} = 7.75$, $p < 0.05$), increased NPP to $4.6 \pm 0.5$ mmol $O_2$
m$^{-2}$ hr$^{-1}$ ($F_{1,15} = 17.19$, $p < 0.05$), and increased GPP to $7.5 \pm 0.5$ mmol $O_2$ m$^{-2}$ hr$^{-1}$ ($F_{1,15} = 26.77$, $p < 0.05$). GPP/R
in T + CM treatments was $1.21 \pm 0.13$, a change which was not significantly different from control chambers



$(F_{1,15} = 3.79, p = 0.075)$. T + CM chambers were net dissolving at night $(-1.8 \pm 0.3$ mmol $CaCO_3$ $m^{-2}$ $hr^{-1})$ and net
calcifying during the day $(2.4 \pm 0.4$ mmol $CaCO_3$ $m^{-2}$ $hr^{-1})$. Overall, 22-hour diel $G_{net}$ in T + CM treatments was
$0.2 \pm 0.2$ mmol $CaCO_3$ $m^{-2}$ $hr^{-1}$, a change which was not significantly different from control chambers $(F_{1,15}$
$= 0.87, p = 0.368)$.
Measured R in all chambers under all treatments was not significantly correlated with NPP $(r = 0.53, df = 31, p$
$> 0.05)$. However, in the C chambers, R was significantly correlated with NPP $(r = 0.81, df = 31, p < 0.05)$.
$CaCO_3$ precipitation during the day was positively correlated with NPP $(r = 0.81, df = 31, p < 0.05;$ slope $= 0.22$
$\pm 0.08)$ while dissolution at night was positively correlated with R $(r = 0.83, df = 31, p < 0.05;$ slope $= 0.45 \pm$
$0.04)$. Average diel $G_{net}$ was positively correlated with GPP/R $(r = 0.83, df = 31, p < 0.05;$ slope $= 0.70 \pm 0.05)$.
The $DIC_{org}:O_2$ quotient for all treatments was $0.94 \pm 0.09$ on average and did not significantly differ from 1 $(p <$
$0.05;$ Fig. 7), suggesting that sulfate reduction did not significantly contribute to the $A_T$ fluxes.
**4. Discussion**
**4.1 The response in coral reef sediment metabolism to seawater warming**
In our experiment, seawater warming $(+2.4 \pm 0.5$ °C) was within the projection of the IPCC RCP 8.5 $(+2.2 - 2.7$
°C). Under this elevated seawater temperature (T), R increased to a greater extent than GPP, shifting the
sediments to net heterotrophy (GPP/R = 0.93) over the 22-hour incubation period (Fig. 8). Whereas NPP and R
were significantly correlated in control chambers $(p < 0.05)$, they were not significantly correlated in the
warming treatments $(p = 0.136)$, evidence that warming decoupled the balance in autotrophic: heterotrophic
metabolism (Fig. 8). The decrease of GPP/R due to warming can be explained by the relatively lower measured
$Q_{10}$ value for GPP $(7.3 \pm 1.2)$ compared to R $(10.7 \pm 3.1)$. These results agree with other studies showing that
seawater warming preferentially enhances R to a greater degree than GPP in marine sediments (Hancke and
Glud, 2004; Weston and Joye, 2005; Tait and Schiel, 2013). The decline in GPP/R in response to warmer
seawater temperature may be a product of the differential ranges in activation energies for GPP and R (Yvon-
Durocher et al., 2010), where R exhibits a stronger and more rapid physiological acclimation to warming
compared to GPP during short-term temperature variations (Wiencke et al., 1993; Robinson, 2000).
The observed 29% decrease in GPP/R in response to warming lead to a net 109% decrease in $G_{net}$ (relative to
control chambers), resulting in a transition to net sediment dissolution over the 22-hour incubation period (Fig.
8). This decrease in $G_{net}$ was most likely due to a respiration-driven increase in porewater $pCO_2$ (e.g., Cyronak et
al., 2013a), thereby decreasing the mean aragonite saturation state in the water column (mean $\Omega_{arg} = -0.7$ relative





to control chambers) and porewater (where sediment dissolution occurred). While increasing T increases $\Omega_{arg}$
geochemically, the biologically driven changes in $\Omega_{arg}$ were most likely the dominant effect on the measured
enhanced dissolution of the sediment given that a 2.4 ℃ increase in temperature would only increase $\Omega_{arg}$
roughly 0.058 units.
Together, our results show that the warming of seawater by 2.4 ℃ will decrease GPP/R 0.38 units and $G_{net}$ 0.2
mmol $CaCO_3$ $m^{-2}$ $hr^{-1}$ in the permeable calcium carbonate sediments at this study site on Heron Island. The
decline in the GPP/R in response to warming implies that a greater fraction of the carbon fixed by autotrophs
was remineralised by heterotrophic bacteria and released as $CO_2$, thus compromising the capacity of coral reef
permeable carbonate sediments to remain net autotrophic at an elevated seawater T. While a transition to net
sediment dissolution under warmer conditions would consume $CO_2$, potentially alleviating some of the $CO_2$
release caused by a transition to net heterotrophy, a comparison of the rates measured in this study show the net
effect would still result in a production of $CO_2$. Under the warmed conditions in this study, organic metabolism
released ~ 5.28 mmol $CO_2$ $m^{-2}$ $d^{-1}$ while inorganic metabolism consumed ~ 1.44 mmol $CO_2$ $m^{-2}$ $d^{-1}$, which
resulted in a net production of 3.84 mmol $CO_2$ $m^{-2}$ $d^{-1}$ in the chambers.
Where the decline in marine sediment GPP/R in response to seawater warming has been previously reported in
several studies (e.g., Woodwell et al., 1998; Hancke and Glud, 2004; Weston and Joye, 2005; Lopez-Urrutia and
Moran, 2007), the decline in $G_{net}$ has only been reported once (Trnovsky et al., 2016). It is important to note that
these results should not be extrapolated beyond 2100, where SST continues above +2.4 ℃. The T increase
simulated in this study (+2.4 ℃) was within the optimal temperature range (30.6 ℃) of previously reported
temperature-metabolism hyperbolic relationships in marine sediments (Yvon-Durocher et al., 2010). Given that
these hyperbolic relationships show that further increases in temperature (+3 - 5 ℃) can have an opposite effect
on sediment metabolism (net decrease in GPP and R; Weston and Joye, 2005), we cannot conclude the results
obtained here would scale linearly beyond ca. 2100.
**4.2 The response in coral reef sediment metabolism to organic matter enrichment**
Increased concentrations of organic matter (OM), analogous to eutrophic conditions on degraded coral reefs,
enhanced both GPP and R in the sediment and likely released nitrogen and phosphorus via organic matter
degradation (ΔGPP/R +0.27 relative to control chambers). These results agree with prior work, where increased
concentrations of OM were quickly aerobically degraded by bacteria (within minutes - see Maher et al., 2013; to
hours - see Ferrier-Pages et al., 2000) and enhanced GPP more than R (Glud et al., 2008; Eyre et al., 2008).





While some of this OM was likely degraded in the water column, previous experiments (e.g., Wild et al., 2004b)
have shown that the high permeability of carbonate sediments permits the transport of OM into the upper
centimetres (1 - 4 cm) of the sand, where bacterial degradation rates can exceed those of the water column by a
factor of 10-12 (Moriarty, 1985; Wilkinson, 1987). Measured changes in GPP and R in response to elevated
concentrations of OM in this study are therefore most likely a product of changes in metabolism in the bacterial
communities residing in the upper layers of the sediment.
Phytodetritus (PD) and coral mucus (CM) enhanced respiration 1.1- and 0.6-fold, respectively, which was a less
pronounced increase in R than the 1.5-fold increase observed by Wild et al. (2004b) using the same *Acropora*
spp. mucus at Heron Island. However this discrepancy may be due to the fact their study used almost three times
more CM (~ 280 ml) per treatment than this study (94 ml). An increase in GPP/R to 1.7 one day following the
deposition of coral spawning material at the same study site (Glud et al., 2008), was similar to the average
increase in GPP/R to 1.6 observed under increased OM concentrations in this study. PD enhanced GPP and R to
a greater degree than CM, which may be explained by the different concentration of nitrogen in each source of
OM. Particulate organic carbon additions differed by less than 10% between PD and CM treatments, whereas
particulate organic nitrogen addition (N) was almost twice as high in the PD compared to the mucus CM, as
indicated by the differing POC:PON ratio for PD (9:1) and CM (16:1). In general, bacterial communities
responsible for the cycling of nutrients in sediments are thought to be nitrogen limited (Eyre et al., 2013). Given
the relatively short timescale (24 hours) in which the response in sediment metabolism to OM was measured, we
reason that the PD was more rapidly mineralized than CM due to a higher N content in the added PD (Oakes et
al., 2011).
To our knowledge, this is the first experiment to examine the short-term relationship between OM degradation
and $G_{net}$ in coral reef sediments. Our results show that increased concentrations of PD and CM both enhanced
$G_{net}$ within the first 22 hours. Most likely the increase in $G_{net}$ was a product of the same biogeochemical
mechanism influencing $G_{net}$ under seawater warming, whereby changes in GPP/R modify porewater $pCO_2$ and
$\Omega_{arg}$. In the case of OM, a preferential enhancement of GPP over R resulted in an increase in $\Omega_{arg}$ (mean $\Omega_{arg}$ =
+0.6 relative to control chambers) and subsequent increase in $G_{net}$ (net precipitation) (+1.4 mmol $CaCO_3$ $m^{-2}$ $hr^{-1}$
relative to control chambers). While the results presented here are the first to report a positive OM-$G_{net}$
relationship specifically in permeable calcium carbonate sediments, a similar response has also been observed at
the coral reef ecosystem level (Yeakel et al., 2015), where increased offshore productivity in the Sargasso Sea
over the course of several months lead to an increase in community $G_{net}$ on the adjacent Bermuda coral reef flat.





Interestingly, this increase in $G_{net}$ in Bermuda coincided with a period of net heterotrophy on the reef. The
difference in the $G_{net}$ – GPP/R relationship between the data in this study (OM increased GPP/R and increased
$G_{net}$) and those in Yeakel et al. (2015) (OM decreased GPP/R and increased $G_{net}$) may be a result of the
timescale of observation. This implies that, should elevated concentrations of OM persist for an extended period
of time (weeks to months), the immediate preferentially phototrophically-mediated recycling of nutrients, and
associated increased GPP/R and $G_{net}$ in coral reef sediments, may eventually shift to net heterotrophy despite the
ability to maintain a positive $G_{net}$.
**4.3 The response in coral reef sediment metabolism to a combination of seawater warming and organic**
**matter enrichment**
The combination of seawater warming and increased concentrations of OM, for both PD and CM, exhibited an
additive enhancement of GPP (+17% relative to the temperature alone) and R (+11% relative to temperature
alone) but countered the effect on GPP/R and $G_{net}$ (no significant difference from the control). Given the effect
of each of these treatments (T and OM) independently on sediment GPP/R and $G_{net}$, and the significant positive
correlation between $G_{net}$ and GPP/R, this result is not surprising. A decrease in GPP/R and $G_{net}$ due to warming
was countered by an increase in GPP/R and $G_{net}$ due to an increased concentration of OM.
This finding raises questions within the context of each treatment, as mean SST on coral reefs will continuously
rise from now until beyond 2100, consistently affecting sediment metabolism. However, organic matter
enrichment of permeable coral reef carbonate sediments is also likely to gradually increase due to enhanced
algal production from elevated nutrients (Furnas et al., 2005), enhanced mucus production due to enhanced
terrestrial sedimentation (Alongi and McKinnon, 2005) and elevated terrestrial input of OM (Diaz-Ortega and
Hernandez-Delgado, 2014). As discussed above this long-term enrichment with OM will most likely make coral
reef sediments more heterotrophic (and not more autotrophic as in this short-term study). However the
subsequent response in $G_{net}$ over longer timescales is less clear, as some work has shown that the degradation of
organic matter can enhance sediment dissolution (Andersson, 2015) whereas other work (e.g., Yeakel et al.,
2015) has shown that community calcification may actually increase. Therefore, combined with an increase in
T, the effect of long-term enrichment of OM on GPP/R is likely to be additive (decrease GPP/R), but the long-
term response in $G_{net}$ still needs to further examination.
Similarly, the effect of other, more persistent products of eutrophication, namely dissolved inorganic nutrients
(DIN: $NH_4^+$, $NO_3^-$, $PO_4^{3-}$), on coral reef sediment GPP/R and $G_{net}$ have yet to be studied and may become more





frequent and persistent as coastal land use changes continue to facilitate the increased runoff of fertilizers (Koop
et al., 2001). Consequently, the results presented here provide an estimation of the future short-term response in
coral reef sediment GPP/R and $G_{net}$ to select forms of global warming (+2.4 °C) and eutrophication (PD and
CM), but by no means have explored other potential warming- and eutrophication-mediated perturbations that
continue to threaten coral reef ecosystems. Future work should consider varying frequencies (e.g., > 24 hours)
and forms of eutrophication (e.g., DIN) as well as a range of T, both within and beyond reported optimal ranges
(> 2.4 °C), to better constrain our understanding of the potential feedback responses in coral reef sediment
GPP/R and $G_{net}$.
**4.4 Conclusions**
Overall, the results of this study suggest that seawater warming will shift GPP/R and $G_{net}$ in permeable calcium
carbonate coral reef sediments to a state of net heterotrophy and net dissolution, respectively, by the year 2100.
Alternatively, short-term eutrophication, and the subsequent production of OM in the form of phytodetritus and
coral mucus, could enhance sediment GPP/R and $G_{net}$. The combined effect of seawater warming and increased
concentrations of OM may additively enhance sediment GPP and R, but the net effect on GPP/R and $G_{net}$ will
likely counter one another on relatively short timescales (22 hours). The future response in the net-flux-
behaviour of $CO_2$ and $O_2$ in the coral reef sediment environment, and the consequent rate of carbon
sequestration into the sediments, will likely depend on the relative frequency of each perturbation. The effects of
OM (e.g., phytoplankton growth, reef-wide mucus secretion) on sediment metabolism generally persist
temporarily (days to weeks) relative to global warming, a constant process which will continue to occur
throughout this century and beyond. Provided this ecological context and the findings from this study, we
propose that increased concentrations of OM, in the form of phytodetritus and coral mucus, will increase $G_{net}$
and GPP/R in the sediment on relatively short timescales. However, once seawater temperature on coral reefs
rises 2.4 °C above the present day mean, the immediate effect of OM on sediment metabolism will be
compromised by a warming-mediated net decrease in $G_{net}$ and GPP/R, thereby limiting the ability of permeable
calcium carbonate sediments on coral reefs to accumulate calcium carbonate.
**Acknowledgements**
We would like to thank Jacob Yeo for his assistance in the field. This research was funded by ARC Discovery
Grant DP150102092.





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



618    **Tables**

**Table 1:** Concentrations of carbon ($\mu$mol C L$^{-1}$) and nitrogen ($\mu$mol N L$^{-1}$) and measured temperature (ºC) in the control and treatment chambers. Values correspond to the mean $\pm$ SE. Control (C) (n = 9) and temperature (T) (n = 7) treatments were pooled together from all four incubations. Organic matter (OM) (phytodetritus (PD) and coral mucus (CM)) and combination treatments (T + PD, T + CM) are pooled together from the two incubations for that specific OM treatment (n = 4).

| Treatment | Carbon ($\mu$mol C L$^{-1}$) | Nitrogen ($\mu$mol N L$^{-1}$) | Temperature (ºC) |
|---|---|---|---|
| C | 0.63 $\pm$ 0.13 | 0.12 $\pm$ 0.08 | 28.2 $\pm$ 1.1 |
| T | 0.63 $\pm$ 0.13 | 0.12 $\pm$ 0.08 | 30.6 $\pm$ 1.0 |
| PD | 21.7 $\pm$ 1.0 | 2.3 $\pm$ 0.8 | 28.4 $\pm$ 1.0 |
| T + PD | 21.7 $\pm$ 1.0 | 2.3 $\pm$ 0.8 | 30.5 $\pm$ 0.9 |
| CM | 24.2 $\pm$ 1.1 | 1.5 $\pm$ 0.4 | 28.3 $\pm$ 0.8 |
| T + CM | 24.2 $\pm$ 1.1 | 1.5 $\pm$ 0.4 | 30.7 $\pm$ 1.1 |



**Table 2:** The equations used in this study to calculate rates of sediment metabolism based on measured

fluxes in dissolved oxygen (DO) and total alkalinity ($A_T$) (Eyre et al. (2011).

| Metabolic Rate | Definition |
|---|---|
| Respiration (R) | Dark DO Flux x -1 |
| Net Primary Production (NPP) | Light DO Flux |
| Gross Primary Production (GPP) | NPP + R |
| GPP/R | GPP x 12 (daylight hours)/ R x 24 (total hours) |
| Net Calcification ($G_{net}$) | $A_T$ Flux x 0.5; positive values represent net calcification and negative rates represent net dissolution |





**Table 3:** Calculated gross primary productivity (GPP: mmol $O_2$ $m^{-2}$ $hr^{-1}$) respiration (R: mmol $O_2$ $m^{-2}$ $hr^{-1}$), the ratio of GPP/R, and net calcification ($G_{net}$: mmol $CaCO_3$ $m^{-2}$ $hr^{-1}$) in the control and treatment chambers. Values correspond to the mean ± SE. Control (C) (n = 9) and temperature (T) (n = 7) treatments were pooled together from all four incubations. OM treatments (phytodetritus (PD) and coral mucus (CM)) and combination treatments (T + PD, T + CM) are pooled together from the two incubations for that specific OM source (n = 4).

| Treatment | R (mmol $O_2$ $m^{-2}$ $hr^{-1}$) | GPP (mmol $O_2$ $m^{-2}$ $hr^{-1}$) | GPP/R | $G_{net}$ (mmol $CaCO_3$ $m^{-2}$ $hr^{-1}$) |
|---|---|---|---|---|
| C | - 1.3 ± 0.5 | 3.2 ± 0.6 | 1.31 ± 0.1 | 0.2 ± 0.2 |
| T | - 3.5 ± 0.4 | 6.4 ± 0.5 | 0.91 ± 0.1 | - 0.1 ± 0.1 |
| PD | - 2.6 ± 0.5 | 7.9 ± 0.4 | 1.54 ± 0.1 | 0.6 ± 0.2 |
| T + PD | - 3.1 ± 0.5 | 7.8 ± 0.5 | 1.27 ± 0.1 | 0.3 ± 0.1 |
| CM | - 2.0 ± 0.4 | 6.4 ± 0.7 | 1.61 ± 0.2 | 0.5 ± 0.2 |
| T + CM | - 2.9 ± 0.4 | 7.4 ± 0.5 | 1.25 ± 0.1 | 0.2 ± 0.2 |





619 **Figures**

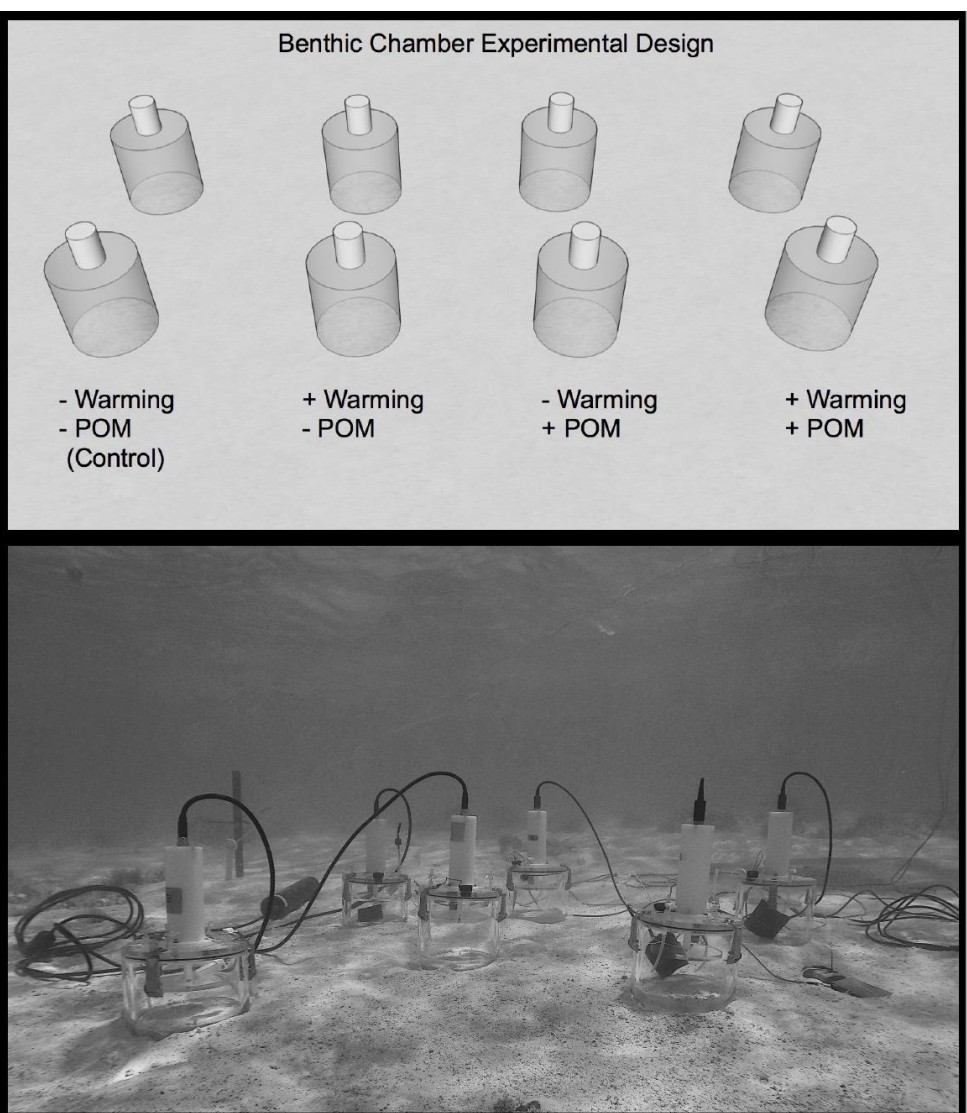

620

621 **Figure 1:** Layout of the experimental design using benthic chambers. Eight chambers were used in total, which

622 provided two replicates per treatment. Chambers are organized by the presence (+) and absence (-) of the

623 warming (+2.4 ºC) and organic matter (OM) (phytodetritus or coral mucus) treatments.





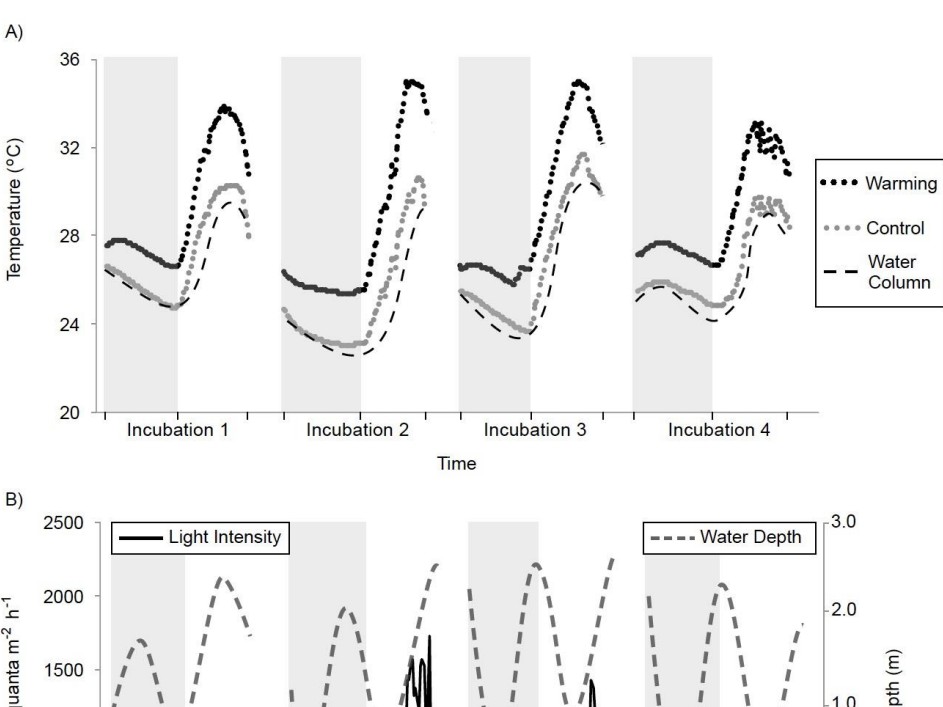

624

**Figure 2:** Water column parameters measured during the four incubations, each starting at sunset (18:00) and

ending at the following day's dusk (16:00). Data are presented from the first phase (Incubation 1 and 2) where

phytodetritus was used as an organic matter (OM) treatment, and from the second phase (Incubation 3 and 4),

where coral mucus was used as an OM treatment. Shaded grey bars represent night time. A) Mean temperature

(°C) measured by Hobo temperature recorders that logged temperature at fifteen-minute intervals during each

incubation period. Data are pooled together as the mean from control (grey dots) and warming (black dots)

treatments (n = 4 per incubation). Mean water column temperature (n = 1 per incubation) shown as a black

dash. B) Measured light intensity ($\mu$mol quanta m$^{-2}$ s$^{-1}$) in the water column (black line) and water height (m)

during each incubation period (grey dash).



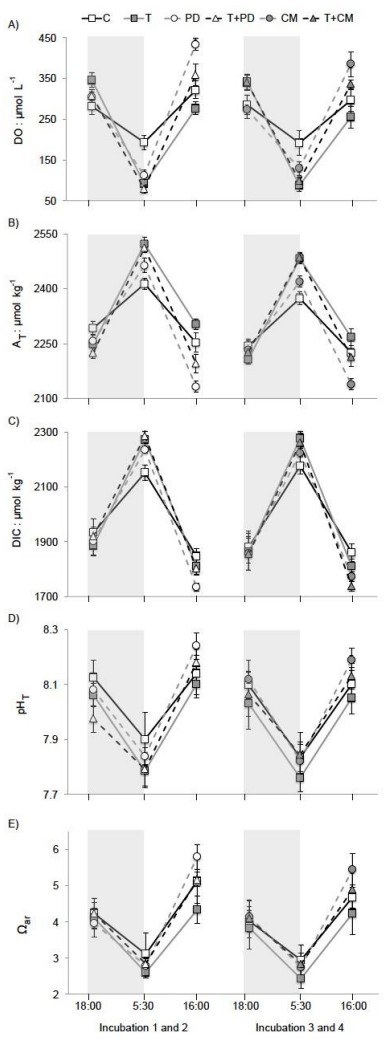

634

**Figure 3:** Water chemistry (mean ± SE) measured and calculated during the four incubations. Control (C),

warming (T), phytodetritus (PD), coral mucus (CM), and combination (T + PD, T + CM) treatments are

averaged over the two incubations (and replicate chambers therein) in which each respective OM treatment was

used (n = 4). Shaded grey bars represent the dark and time of sampling is labelled on the x-axis. A) Measured

fluxes in dissolved oxygen (DO: $\mu mol\ L^{-1}$). B) Measured fluxes in total alkalinity ($A_T$: $\mu mol\ kg^{-1}$). C) Measured

fluxes in dissolved inorganic carbon (DIC: $\mu mol\ kg^{-1}$). D) Calculated changes in pH (total scale: $pH_T$). E)

Calculated fluxes in aragonite saturation state ($\Omega_{ar}$).





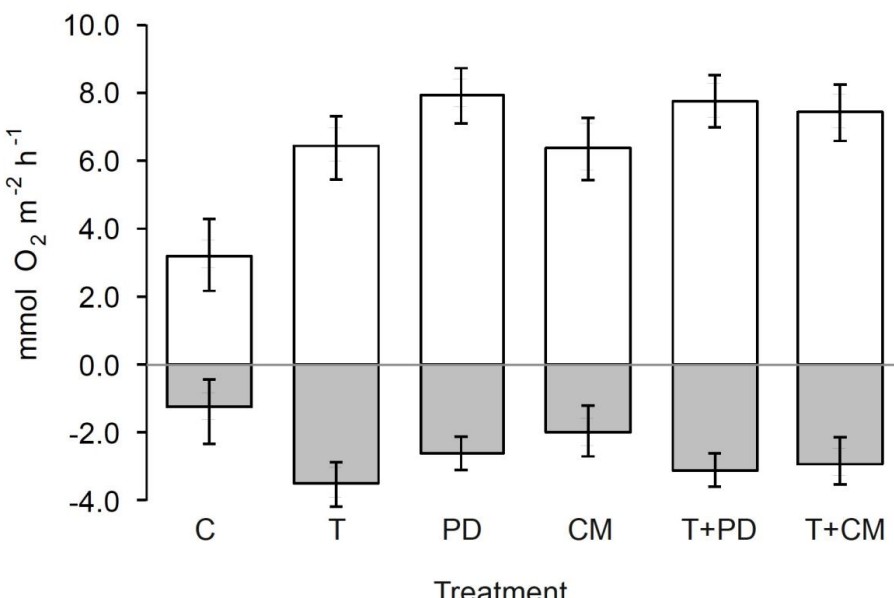

642

**Figure 4:** Mean sediment gross primary production (GPP: mmol $O_2$ m$^{-2}$ h$^{-1}$) and respiration (R: mmol $O_2$ m$^{-2}$ h$^{-1}$) in response to warming (+2.4 °C) and each OM treatment (phytodetritus and coral mucus). Control (C) (n = 9) and warming (T) (n = 7) treatments are averaged over all four incubations and the replicate chambers therein. Phytodetritus (PD), coral mucus (CM), and combination (T + PD, T + CM) treatments are averaged over the two incubations (and replicate chambers therein) in which each respective OM treatment was used (n = 4). Average measured rates ± SE are represented in white for GPP (positive) and grey for R (negative).




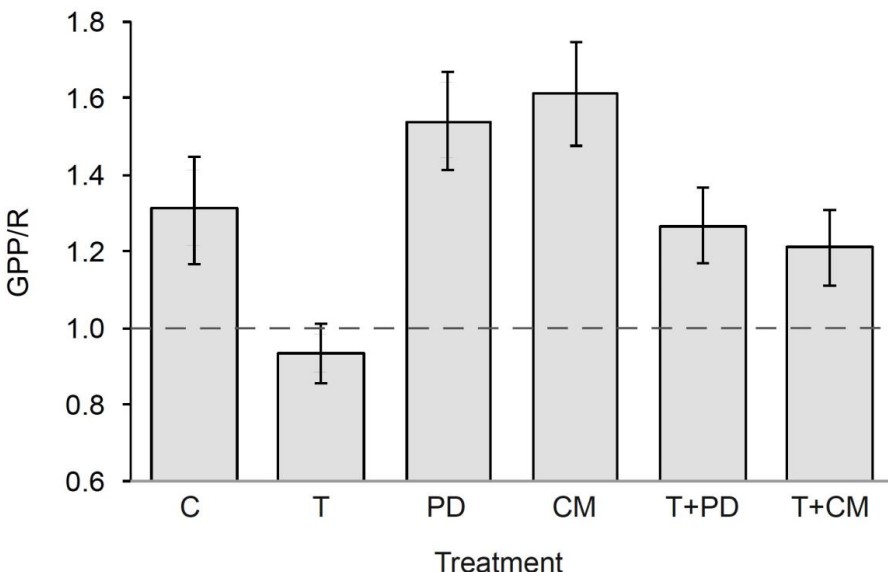

649

**Figure 5:** Mean ± SE sediment gross primary production (12 hour) to respiration (24 hour) ratios (GPP/R) in

response to warming (+2.4 °C) and each OM treatment (phytodetritus and coral mucus). Control (C) (n = 9) and

warming (T) (n = 7) treatments are averaged over all four incubations and the replicate chambers therein, while

phytodetritus (PD), coral mucus (CM), and combination (T + PD, T + CM) treatments are averaged over the two

incubations (and replicate chambers therein) in which each respective OM treatment was used (n = 4). Dashed

grey line represents the divide between net heterotrophy and net autotrophy (GPP/R = 1).



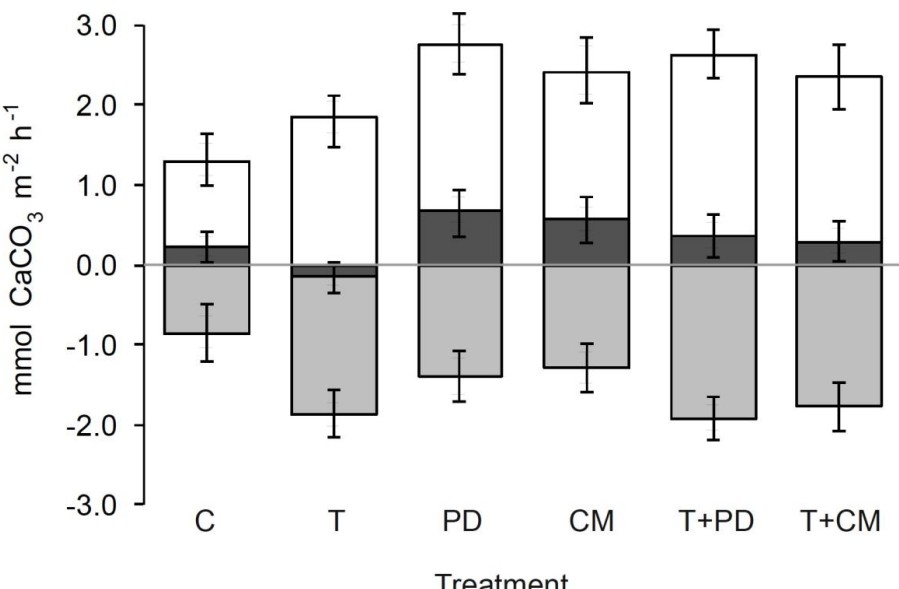

656

**Figure 6:** Mean sediment net calcification ($G_{net}$: mmol $CaCO_3$ m$^{-2}$ h$^{-1}$) in response to warming (+2.4 °C) and each OM treatment (phytodetritus and coral mucus). Control (C) (n = 9) and warming (T) (n = 7) treatments are averaged over all four incubations and the replicate chambers therein, while phytodetritus (PD), coral mucus (CM), and combination (T + PD, T + CM) treatments are averaged over the two incubations (and replicate chambers therein) in which each respective OM treatment was used (n = 4). Average measured rates ± SE are represented in white for light $G_{net}$ (positive) and grey for dark $G_{net}$ (negative). Black bars represent the 24-hour diel $G_{net}$ averaged from light and dark measurements.



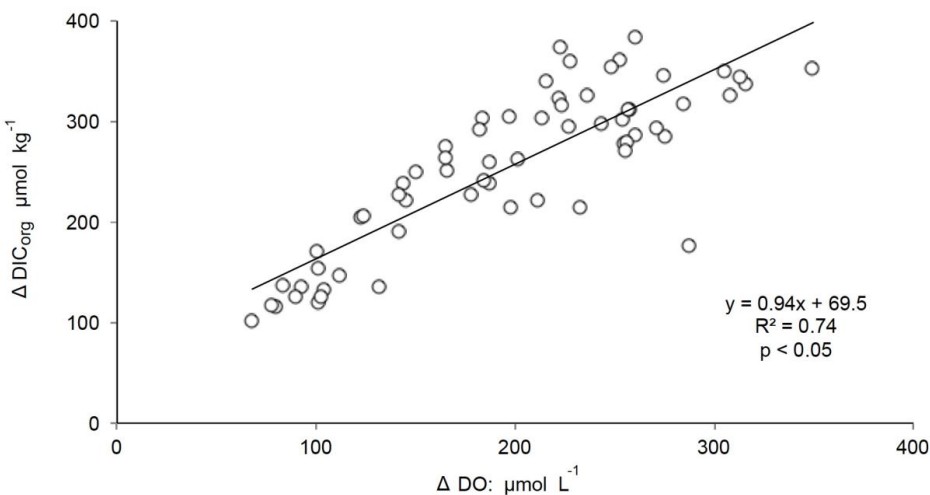

664

**Figure 7:** A linear correlation between calculated changes in dissolved inorganic carbon ($\Delta DIC_{org}$: $\mu mol\ kg^{-1}$) as

a function of measured changes in dissolved oxygen ($\Delta DO$: $\mu mol\ L^{-1}$) over each 12-hour sampling period from

all chambers and incubations. To examine the variation in DIC due solely to photosynthesis and respiration

($DIC_{org}$), changes in DIC were corrected for calcium carbonate precipitation/dissolution using the measured

changes in total alkalinity ($A_T$) (0.5 moles $CO_2$: 1 mole $A_T$).




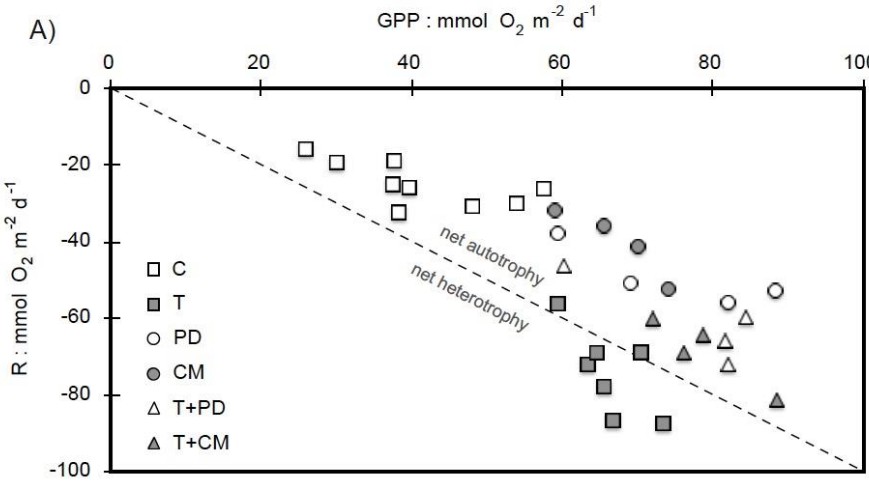

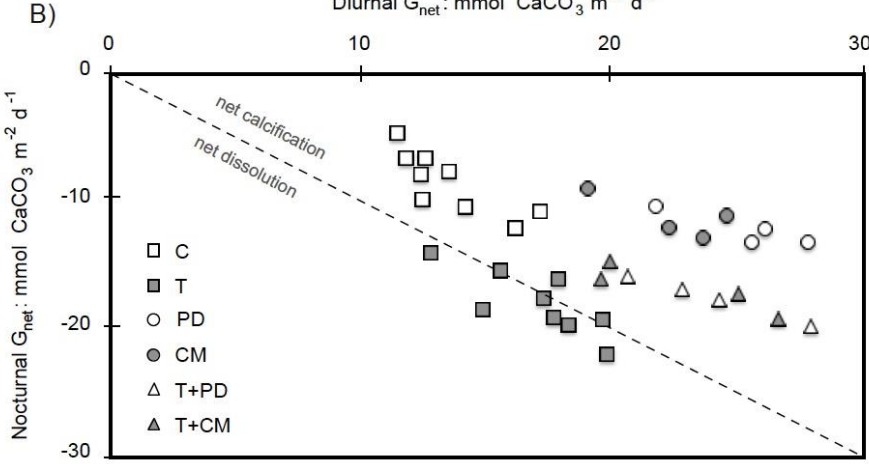

670

**Figure 8:** Measured metabolic rates from the control (C) (n = 9) and warming (T) (n = 7) treatments are displayed from all four incubations and the replicate chambers therein. Phytodetritus (PD), coral mucus (CM), and combination (T + PD, T + CM) treatments are displayed from the two incubations (and replicate chambers therein) where each respective OM treatment was used (n = 4). A) Respiration (R: mmol $O_2$ m$^{-2}$ d$^{-1}$) plotted as a function of gross primary production (GPP: mmol $O_2$ m$^{-2}$ d$^{-1}$). Dashed line represents the divide between net heterotrophy and net autotrophy (GPP/R = 1). B) Dark dissolution (Dark G: mmol $CaCO_3$ m$^{-2}$ d$^{-1}$) plotted as a function of daytime calcification (Diurnal G: mmol $CaCO_3$ m$^{-2}$ d$^{-1}$). Dashed line represents the divide between net calcification and net dissolution ($G_{net}$ = 0).