# Peer review of "The short-term combined effects of temperature and organic matter enrichment on permeable coral reef carbonate sediment metabolism and dissolution"

_Biogeosciences, 2017_

## Referee Comment (RC1) · Anonymous Referee #1 · 22 Jul 2017

This paper makes some important contributions to the topic of how permeable carbonate sediments in a coral reef setting will response to a 2.4 C warming and to organic matter enrichment. The experiment was well designed, executed and adequately replicated. They found that the sediments were undergoing net dissolution during the night time hours under control and all treatment conditions despite the fact that the overlying water in the chambers was supersaturated with respect to aragonite (omega ar = 2.5-4.0). This alone is noteworthy. It has been reported in field studies but it is helpful to confirm the observation under well constrained and replicated experimental conditions.

[Figure]

It is also interesting that during the daylight period they observed net carbonate precipitation under control and all treatment conditions. The authors should be encouraged to comment on what they think is contributing to this carbonate production. Forams possibly?

The main findings of the study are that elevated temperature (+2.4C) caused both R and GPP to increase by significant amounts. R increased more than GPP so that the GPP/R went for 1.3 to 0.9, i.e. from net autotrophic to net heterotrophic. This a reasonable result with many previous studies finding the dark respiration being more sensitive to temperature than photosynthesis. The Q10s for R and GPP are extremely high at 10.7 and 7.3, respectively. The authors need to discuss these results and put them in the context of the literature. Typically Q10 values are in the 2.0 to 2.5 range and this is consistent with the energy of activation for enzymatically mediated reactions which underlies the theory of why the rates are temperature dependent. Q10s are best computed on a C-specific basis, i.e. grams of C fixed or respired per gram C of organism biomass. I am not sure that a Q10 computed from R and GPP normalized to substrate area is meaningful. These high values are suggesting that something more than just a temperature effect on the energy of activation of the biological processes is at work. I think it would be better to simply report the temperature sensitivity on a mmol/m2/h per degree C basis and not suggest that that dependence might hold over a broader temperature range until there is data to support the claim. The reported effect of the temperature increase on Gnet varies between Table 3 and the text and this needs to be resolved. Table 3 says that Gnet is 0.2+/-0.2 mmol/m2/h under control conditions and -0.1+/-0.1 under the elevated temperature treatment. In the text Gnet under elevated temperature is said to be -0.2+/-0.1. It is hypothesized that the shift from net carbonate precipitation to net dissolution on a daily basis is caused by the shift in organic carbon metabolism from net autotrophic to net heterotrophic. This is supported by the observation that omega arag is lowest at dawn in the T treatments. The authors cite Yeakel et al in support of the connection between net heterotrophy and dissolution. It would be relevant to cite Muehllehner et al 2016 as another study

that reported a clear relationship between reef sediment dissolution and a seasonal shift between community autotrophy and heterotrophy.

The observed responses to organic matter enrichment are among the most interesting of this study. They observed that the PD and CM enrichments resulted in increases in R and GPP, although the increase in GPP was greater than the increase in R. The effect of the organic matter enrichment also overwhelmed the effect of temperature such the GPP/R and Gnet were not significantly different from the control. The authors suggest that what happened is that first the organic matter was remineralized to its nutrient constituents. The small increase in R would be consistent with this. Then the released nutrients were immediately taken up of the autotrophs in the system resulting in the observed increase in GPP. The net autotrophy would result in a small elevation in pH which would in turn bump up saturation state and account for the shift from net dissolution to net carbonate precipitation. This scenario is reasonable to me. What is very interesting is that the system seems to be very closely poised at a tipping point. Day-night shifts in pH and temperature and organic matter augmentation are all able to shift the pore water saturation state sufficiently to shift the system between net carbonate production or dissolution.

I would encourage the authors to include a table where they compare their daily rates of carbonate production and dissolution with the rates reported in the literature for other locations.

As a small technical detail it would be nice if the authors employed the letter system to indicate in figures 4-6 which means are significantly different and which are not. The information can be obtained from the text but the figures would be more useful if the information was also supplied there.

---

## Referee Comment (RC2) · A. Hannides (Referee) · 2 Aug 2017

The manuscript describes a study that falls into a now well-established tradition of permeable sediment experimental studies at Heron Reef. I think that it complements previous findings very well, by extending our fundamental knowledge of how these sands work and are expected to respond in view of future change. Surprisingly, despite their preponderance, sands do not receive more attention and remain understudied. In view of the above, I find this manuscript worthy of publication in this journal.

[Figure]

The study is justified by a substantial review, the experiments are well designed and sufficient to support the scope of the study and to reach the stated conclusions. However, there are some aspects of the manuscript that need improvement before it is published.

One important correction to be made involves the recipes for organic matter addition, described in section 2.5, "Organic matter manipulations." The math doesn't add up. On p. 7, line 178, the phytodetritus concentrate is characterized by concentrations of 8.5 umol C/L and 0.9 umol N/L, and we are then told that when 10 mL are added to ∼4 L and diluted, the concentrations of C and N almost triple! Could the mentioned units be actually mmol instead of umol? On p. 8, line 191, we are told that 94 mL of mucus were added to corresponding treatments. At concentrations of 12.1 mmol C/L and 0.8 mmol N/L (line 194), dilution by 4 L of overlying water would yield 280 umol C/L or roughly 10 times higher than those in Table 1. Please re-examine these recipes and correct accordingly.

Another important aspect of the study that needs improvement is the description of statistical analyses to test the proposed hypotheses and their outcomes. Currently, statistical aspects of the study are spread far and wide in the text, tables and figures, and are occasionally redundant. Below are some suggestions for improvement.

A statement like the following is repeated in the legend of several tables and figures: "Values correspond to the mean ± SE. Control (C) (n = 9) and temperature (T) (n = 7) treatments were pooled together from all four incubations. Organic matter (OM) (phytodetritus (PD) and coral mucus (CM)) and combination treatments (T + PD, T + CM) are pooled together from the two incubations for that specific OM treatment (n = 4)." Mention this pooling strategy once in Methods, and that should be sufficient. This should unclutter a lot of the legends. If you so wish, include values of n in the treatment column of Tables 1 and 3 in parentheses.

The abstract states that "The combined effect of warming and OM enhanced R and

GPP, but the net effect on GPP/R and Gnet was not significantly different from control incubations." A simple and important statement like this cannot be verified easily. Sure, the bar charts showing means and standard errors can be visually inspected and the statement (kind of) verified, but the statistical proof is buried in the text. One way to resolve this is to use symbols on bar charts (Figures 4, 5 and 6) to indicate statistically insignificantly different treatments, i.e., same symbol indicates indistinguishable values.

The Results section is festooned with statistic and probability values in parentheses. Consider displaying all results of your ANOVA tests in a table to precede or follow Table 3 and focus your Results section on highlighting the main outcomes. In my opinion, the readability problem in this section is exacerbated by a tendency to repeat values for T, GPP, R, GPP/R etc. already shown in Table 3 and the figures. There's no need to repeat these values; just refer to Table 3 and the relevant figure.

A final comment on the statistics front concerns the use of a Model I regression "to fit a linear relationship for the purpose of predicting inorganic metabolism (Gnet) from organic metabolism (NPP, GPP, R, GPP/R)." Since the latter are not true independent variables, a Model II regression may be the appropriate approach towards this goal.

Beyond the two major topics I mentioned above (organic enrichment recipes and statistics), I'd like to make a few more minor suggestions to improve this manuscript.

The excellent overview of past experiments (starting on p. 4) distinguishes between "short" and "long" experiments. It would be useful if the actual time-scales are mentioned explicitly (instead of "hours to days") so that those studies and the one described in the manuscript can be placed in perspective.

The "Sediment grain size: 12.1%. 2 mm ..." statement (p. 5, lines 120-122) is awkward, not even a complete sentence. Is this information important? I think so. Please place it in a table on characteristics of the sand used, and include some basic sediment grain-size statistics (mean and median size, sorting) as well as permeability and porosity.

The "best of three" approach (p. 8, line 209) is too generic a term. Please define it and/or provide a reference.

A semantic point regarding the definition of Respiration, R. I definitely understand why it is elegant to present the magnitude of R as a negative for the purposes of Figure 4. However, R values can be listed as positive values in Table 3, so that the positive GPP/R values make sense. Alternatively, modify the definition of R on p. 9, line 235, as flux across the sediment-water interface, where a negative value indicates flux into the sediment.

Finally, please consider adding two columns in Table 3 after Gnet, to show Gnet night and day values.

---

## Author Comment (AC1) · 19 Aug 2017

General Comments:

This paper makes some important contributions to the topic of how permeable carbonate sediments in a coral reef setting will response to a 2.4 C warming and to organic

matter enrichment. The experiment was well designed, executed and adequately replicated. They found that the sediments were undergoing net dissolution during the night time hours under control and all treatment conditions despite the fact that the overlying water in the chambers was supersaturated with respect to aragonite (omega ar = 2.5-4.0). This alone is noteworthy. It has been reported in field studies but it is helpful to confirm the observation under well constrained and replicated experimental conditions.

Response to General Comments:

We thank the reviewer for their detailed analysis of this manuscript. We agree that the continued compilation of data, such as the results contained herein, are helpful in shaping an ever-evolving understanding of coral reef permeable sediment carbonate chemistry. We have done our best to accommodate each comment and feel that the manuscript benefits from their suggestions. Please note, the referenced line numbers for each comment response refer to a new, revised version of this manuscript and may differ from the older version.

Specific Comments

Comment 1: It is also interesting that during the daylight period they observed net carbonate precipitation under control and all treatment conditions. The authors should be encouraged to comment on what they think is contributing to this carbonate production. Forams possibly?

Response 1: We agree with the reviewer that it is interesting the sediments exhibited net diurnal calcification under all treatment conditions. We have added some discussion as to why such behaviour may have been observed. (Lines 316 – 325)

Comment 2: The main findings of the study are that elevated temperature (+2.4C) caused both R and GPP to increase by significant amounts. R increased more than GPP so that the GPP/R went for 1.3 to 0.9, i.e. from net autotrophic to net heterotrophic. This a reasonable result with many previous studies finding the dark respi-
ration being more sensitive to temperature than photosynthesis. The Q10s for R and GPP are extremely high at 10.7 and 7.3, respectively. The authors need to discuss these results and put them in the context of the literature. Typically Q10 values are in the 2.0 to 2.5 range and this is consistent with the energy of activation for enzymatically mediated reactions which underlies the theory of why the rates are temperature dependent. Q10s are best computed on a C-specific basis, i.e. grams of C fixed or respired per gram C of organism biomass. I am not sure that a Q10 computed from R and GPP normalized to substrate area is meaningful. These high values are suggesting that something more than just a temperature effect on the energy of activation of the biological processes is at work. I think it would be better to simply report the temperature sensitivity on a mmol/m2/h per degree C basis and not suggest that that dependence might hold over a broader temperature range until there is data to support the claim.

Response 2: We thank the reviewer for their detailed analysis of the Q10 values in this manuscript and agree that the presented values are likely meaningless when normalized to substrate area. We agree with the suggested alternate approach and have therefore removed mention of Q10 calculations. In place, we have instead reported the temperature sensitivity on a mmol/m2/d per deg C basis in the methods and results. Please note this metric has been extrapolated to a total diel value over 24 hours (d-1) to provide explanative value for GPP/R in the discussion. (Lines 245, 288, 330)

Comment 3: The reported effect of the temperature increase on Gnet varies between Table 3 and the text and this needs to be resolved. Table 3 says that Gnet is 0.2+/-0.2 mmol/m2/h under control conditions and -0.1+/-0.1 under the elevated temperature treatment. In the text Gnet under elevated temperature is said to be -0.2+/-0.1.

Response 3: We thank the reviewer for their detailed overview of the results. The actual value for Gnet under elevated temperature is -0.15. To provide consistency, both Table 3 and the text will be rounded up to list the value as -0.2+/-0.1.

Comment 4: It is hypothesized that the shift from net carbonate precipitation to net dissolution on a daily basis is caused by the shift in organic carbon metabolism from net autotrophic to net heterotrophic. This is supported by the observation that omega arag is lowest at dawn in the T treatments. The authors cite Yeakel et al in support of the connection between net heterotrophy and dissolution. It would be relevant to cite Muehllehner et al 2016 as another study that reported a clear relationship between reef sediment dissolution and a seasonal shift between community autotrophy and heterotrophy.

Response 4: We thank the reviewer for this valuable additional citation. Muehllehner et al. 2016 has been added to the portion of the introduction where the coinciding seasonal shift to net respiration and dissolution is discussed. (Line 94)

Comment 5: The observed responses to organic matter enrichment are among the most interesting of this study. They observed that the PD and CM enrichments resulted in increases in R and GPP, although the increase in GPP was greater than the increase in R. The effect of the organic matter enrichment also overwhelmed the effect of temperature such the GPP/R and Gnet were not significantly different from the control. The authors suggest that what happened is that first the organic matter was remineralized to its nutrient constituents. The small increase in R would be consistent with this. Then the released nutrients were immediately taken up of the autotrophs in the system resulting in the observed increase in GPP. The net autotrophy would result in a small elevation in pH which would in turn bump up saturation state and account for the shift from net dissolution to net carbonate precipitation. This scenario is reasonable to me. What is very interesting is that the system seems to be very closely poised at a tipping point. Day-night shifts in pH and temperature and organic matter augmentation are all able to shift the pore water saturation state sufficiently to shift the system between net carbonate production or dissolution. I would encourage the authors to include a table where they compare their daily rates of carbonate production and dissolution with the rates reported in the literature for other locations.

Response 5: We agree with the reviewer's synopsis regarding the mechanisms behind the observed trends in Gnet in response to organic matter enrichment and thank them for their detailed interpretation. We further agree that a comparison of carbonate sediment production and dissolution would be valuable in table form. It should be noted that the methodology employed and simulated advection rate varies greatly among past studies, therefore making comparisons amongst all described historical rates problematic. We would direct the reviewer and reader to consult the review paper in Nature Climate Change by Eyre et al. (2014) where these variations in methodologies and subsequent carbonate production and calcification rates are discussed in greater detail. Nevertheless, we have inserted a table into the discussion (Table 4) comparing studies that have specifically employed the same chamber methodology at the same simulated advection rate (sediment percolation rate $\sim$ 43 L m-2 d-1).

Comment 6: As a small technical detail it would be nice if the authors employed the letter system to indicate in figures 4-6 which means are significantly different and which are not. The information can be obtained from the text but the figures would be more useful if the information was also supplied there.

Response 6: We agree that such a notation would be valuable to indicate which means are significantly different from the control. When using the letter system to indicate significant difference between treatments, the figures quickly become crowded with information. For this reason, we have used an asterisk (*) notation to only indicate which means (GPP/R and 24-hour Gnet) were significantly different from the control. This was not necessary for Figure 4, as all treatments were significantly different for both GPP and R, but was necessary in Figure 5 and 6, where variations in significance existed. We feel this does an adequate job of satisfying the reviewer's request while maintaining a clear and informative figure.

Please also note the supplement to this comment:
https://www.biogeosciences-discuss.net/bg-2017-109/bg-2017-109-AC1-

supplement.pdf

[Figure]

[Figure]

**Fig. 1.** Amended Figure 5

**Fig. 2.** Amended Figure 6

---

## Author Comment (AC2) · 19 Aug 2017

A. Hannides (Referee #2)

General Comments:

The manuscript describes a study that falls into a now well-established tradition of permeable sediment experimental studies at Heron Reef. I think that it complements

previous findings very well, by extending our fundamental knowledge of how these sands work and are expected to respond in view of future change. Surprisingly, despite their preponderance, sands do not receive more attention and remain understudied. In view of the above, I find this manuscript worthy of publication in this journal. The study is justified by a substantial review, the experiments are well designed and sufficient to support the scope of the study and to reach the stated conclusions. However, there are some aspects of the manuscript that need improvement before it is published.

Response to General Comments: We thank A. Hannides for a thorough review of the manuscript and insightful comments. We have done our best to address each individual comment and feel the manuscript benefits greatly from these edits. Please note, the referenced line numbers for each comment response refer to a new, revised version of this manuscript and may differ from the older version.

Specific Comments

Comment 1: One important correction to be made involves the recipes for organic matter addition, described in section 2.5, "Organic matter manipulations." The math doesn't add up. On p. 7, line 178, the phytodetritus concentrate is characterized by concentrations of 8.5 umol C/L and 0.9 umol N/L, and we are then told that when 10 mL are added to _4 L and diluted, the concentrations of C and N almost triple! Could the mentioned units be actually mmol instead of umol? On p. 8, line 191, we are told that 94 mL of mucus were added to corresponding treatments. At concentrations of 12.1 mmol C/L and 0.8 mmol N/L (line 194), dilution by 4 L of overlying water would yield 280 umol C/L or roughly 10 times higher than those in Table 1. Please re-examine these recipes and correct accordingly.

Response 1: We thank the reviewer for their detailed analysis of the organic matter manipulations and the expected concentrations. We apologize for the confusion, but the listed concentrations for phytodetritus (8.5 umol C/L and 0.9 umol N/L) indicate the final concentrations in 1 L of seawater if 1 ml of the PD concentrate is added (the

volume filtered). So when 10 ml (10/1 = 10x) are added to 4L of seawater (10/4 = 2.5x) this is why it seems the value triples. Likewise, the coral mucus concentrations (12.1 umol C/L and 0.8 umol N/L) refer to final concentrations in 1 L of seawater if 12 ml of the CM concentrate is added (the volume filtered). So when 94 mL (94/12 = 7.8x) are added to 4L of seawater ((7.8/4 = 1.95x umol/L) the value is almost doubled. This information has now been added to section 2.5 to clarify. (Lines 184, 200)

Comment 2: Another important aspect of the study that needs improvement is the description of statistical analyses to test the proposed hypotheses and their outcomes. Currently, statistical aspects of the study are spread far and wide in the text, tables and figures, and are occasionally redundant. Below are some suggestions for improvement. A statement like the following is repeated in the legend of several tables and figures: "Values correspond to the mean $\pm$ SE. Control (C) (n = 9) and temperature (T) (n = 7) treatments were pooled together from all four incubations. Organic matter (OM) (phytodetritus (PD) and coral mucus (CM)) and combination treatments (T + PD, T + CM) are pooled together from the two incubations for that specific OM treatment (n = 4)." Mention this pooling strategy once in Methods, and that should be sufficient. This should unclutter a lot of the legends. If you so wish, include values of n in the treatment column of Tables 1 and 3 in parentheses.

Response 2: We understand the thorough explanation of values, pooling practices, and sample size can seem redundant. The pooling strategy has now been limited to the methods with the included assumption that values, where pooled together, were not significantly different between incubations. Figure and table legends have been reduced in statistical text to be less redundant.

Comment 3: The abstract states that "The combined effect of warming and OM enhanced R and GPP, but the net effect on GPP/R and Gnet was not significantly different from control incubations." A simple and important statement like this cannot be verified easily. Sure, the bar charts showing means and standard errors can be visually inspected and the statement (kind of) verified, but the statistical proof is buried in

the text. One way to resolve this is to use symbols on bar charts (Figures 4, 5 and 6) to indicate statistically insignificantly different treatments, i.e., same symbol indicates indistinguishable values.

Response 3: A similar comment was posted by the other reviewer, so we agree an amendment is necessary. To meet both requests and keep the figures uncluttered, a * has been added above mean diel Gnet and GPP/R values that significantly differed from the control.

Comment 4: The Results section is festooned with statistic and probability values in parentheses. Consider displaying all results of your ANOVA tests in a table to precede or follow Table 3 and focus your Results section on highlighting the main outcomes. In my opinion, the readability problem in this section is exacerbated by a tendency to repeat values for T, GPP, R, GPP/R etc. already shown in Table 3 and the figures. There's no need to repeat these values; just refer to Table 3 and the relevant figure.

Response 4: We agree that the results section's readability could be improved by removing redundant information. However, if these requests are met (including Comment 10), and all results and statistics are moved to a table, we feel that it becomes the case the manuscript contains too much data in table form and too little in text form. To strike a balance where both the text and the tables provide non-redundant information, we have removed mention of the actual values of each metabolic rate in the text and left this to be consulted in the tables. In turn, the statistical trends and probabilities have remained in the result text so the reader can understand if the trend was an increase or decrease and if this trend was significant. (Lines 276-313)

Comment 5: A final comment on the statistics front concerns the use of a Model I regression "to fit a linear relationship for the purpose of predicting inorganic metabolism (Gnet) from organic metabolism (NPP, GPP, R, GPP/R)." Since the latter are not true independent variables, a Model II regression may be the appropriate approach towards this goal.

Response 5: We thank the reviewer for this notification and understand where a Model II regression would be useful. Upon conducting this analysis, many of the results do not necessarily provide additional explanatory value beyond the already presented significant and non-significant ANOVA results. For this reason, this portion of the manuscript has been removed, as we do not feel it provides additional valuable information to the reader or interpretive advantages not provided by already listed data.

Comment 6: The excellent overview of past experiments (starting on p. 4) distinguishes between "short" and "long" experiments. It would be useful if the actual time-scales are mentioned explicitly (instead of "hours to days") so that those studies and the one described in the manuscript can be placed in perspective.

Response 6: We understand the need for a more specific definition of short and long term as it relates to each study and have attempted to do so in this portion of the introduction with specific mentions of each cited study's duration of measurement. (Lines 80 – 93)

Comment 7: The "Sediment grain size: 12.1%. 2 mm . . ." statement (p. 5, lines 120-122) is awkward, not even a complete sentence. Is this information important? I think so. Please place it in a table on characteristics of the sand used, and include some basic sediment grain-size statistics (mean and median size, sorting) as well as permeability and porosity.

Response 7: We thank the reviewer for noticing this grammatical error. We agree with the need to refine this statement to a more complete sentence and have done so. This manuscript has been formatted to follow the literature from which these measurements were taken (Cyronak et al. 2013b). For a more detailed understanding of the Heron Island sediment characteristics, we direct the reader to Glud et al. (2008), and Cyronak et al. (2013a, 2013b) (Line 126). If the reviewer believes a table is absolutely necessary, one can be added with this data, but we felt it best to first point out this is cited data from previous published research.

Comment 8: The "best of three" approach (p. 8, line 209) is too generic a term. Please define it and/or provide a reference.

Response 8: We understand where this explanation suffers from colloquialism. We have instead phrased the sentence to read "the mean of the three most similar values" and provided a reference (Dickson, 2007). (Line 214)

Comment 9: A semantic point regarding the definition of Respiration, R. I definitely understand why it is elegant to present the magnitude of R as a negative for the purposes of Figure 4. However, R values can be listed as positive values in Table 3, so that the positive GPP/R values make sense. Alternatively, modify the definition of R on p. 9, line 235, as flux across the sediment-water interface, where a negative value indicates flux into the sediment.

Response 9: We understand how this may create confusion for the reader and have added the following text to the methods on p. 9, line 235: "Both NPP and GPP are reported as positive values to represent flux of O2 from the sediment into the chamber water column whereas R is reported as a negative value to represent the flux of O2 from the chamber water column into the sediment. To calculate the ratio of GPP/R, absolute values of R were used." (Lines 243-245)

Comment 10: Finally, please consider adding two columns in Table 3 after Gnet, to show Gnet night and day values.

Response 10: This has been added to Table 3 and removed from the text, thank you. We have also removed NPP values from the text and added these to Table 3 as a means to address the readability issue mentioned in Comment 4.

Please also note the supplement to this comment:
https://www.biogeosciences-discuss.net/bg-2017-109/bg-2017-109-AC2-supplement.pdf

[Figure]

**Fig. 1.** Amended Figure 5

[Figure]

**Fig. 2.** Amended Figure 6

**Table 3:** Calculated respiration (R: mmol $O_2$ m$^{-2}$ hr$^{-1}$), net primary productivity (NPP: mmol $O_2$ m$^{-2}$ hr$^{-1}$), gross primary productivity (GPP: mmol $O_2$ m$^{-2}$ hr$^{-1}$), the ratio of GPP/R, and net calcification ($G_{net}$: mmol $CaCO_3$ m$^{-2}$ hr$^{-1}$) in the control and treatment chambers. Values correspond to the mean ± SE.

| Treatment | R (mmol $O_2$ m$^{-2}$ hr$^{-1}$) | NPP (mmol $O_2$ m$^{-2}$ hr$^{-1}$) | GPP (mmol $O_2$ m$^{-2}$ hr$^{-1}$) | GPP/R | Day $G_{net}$ (mmol $CaCO_3$ m$^{-2}$ hr$^{-1}$) | Night $G_{net}$ (mmol $CaCO_3$ m$^{-2}$ hr$^{-1}$) | Diel $G_{net}$ (mmol $CaCO_3$ m$^{-2}$ hr$^{-1}$) |
|---|---|---|---|---|---|---|---|
| C | - 1.3 ± 0.5 | 1.9 ± 0.3 | 3.2 ± 0.4 | 1.31 ± 0.1 | 1.3 ± 0.2 | - 0.9 ± 0.2 | 0.2 ± 0.2 |
| T | - 3.5 ± 0.4 | 2.9 ± 0.4 | 6.4 ± 0.5 | 0.91 ± 0.1 | 1.7 ± 0.2 | - 1.9 ± 0.2 | - 0.2 ± 0.1 |
| PD | - 2.6 ± 0.5 | 5.3 ± 0.5 | 7.9 ± 0.4 | 1.54 ± 0.1 | 2.8 ± 0.3 | - 1.5 ± 0.2 | 0.6 ± 0.2 |
| T + PD | - 3.1 ± 0.5 | 4.7 ± 0.5 | 7.8 ± 0.5 | 1.27 ± 0.1 | 2.6 ± 0.3 | - 1.9 ± 0.2 | 0.3 ± 0.1 |
| CM | - 2.0 ± 0.4 | 4.4 ± 0.4 | 6.4 ± 0.7 | 1.61 ± 0.2 | 2.4 ± 0.3 | - 1.3 ± 0.2 | 0.5 ± 0.2 |
| T + CM | - 2.9 ± 0.4 | 4.6 ± 0.5 | 7.4 ± 0.5 | 1.25 ± 0.1 | 2.3 ± 0.4 | - 1.8 ± 0.3 | 0.2 ± 0.2 |

**Fig. 3.** Amended Table 3